# Associations of Gender Dissatisfaction with Adolescent Mental Distress and Sexual Victimization

**DOI:** 10.3390/children9081221

**Published:** 2022-08-12

**Authors:** Qiguo Lian, Xiayun Zuo, Chunyan Yu, Chaohua Lou, Xiaowen Tu, Weijin Zhou

**Affiliations:** NHC Key Laboratory of Reproduction Regulation, Shanghai Institute for Biomedical and Pharmaceutical Technologies, Shanghai 200237, China

**Keywords:** gender dissatisfaction, mental distress, sexual victimization, adolescents

## Abstract

Gender dissatisfaction is often linked to adverse health outcomes and is an under-researched area of adolescent health. The aim of our study was to examine the associations of gender dissatisfaction with adolescent mental distress and sexual victimization. We conducted a cross-sectional study in April 2019 using a computerized self-administered questionnaire to collect data on the gender dissatisfaction, mental distress, and sexual victimization among Chinese adolescents. We used multivariate logistic models to estimate sex-stratified adjusted odds ratios (AORs) and 95% confidence intervals (CIs) for the associations of gender dissatisfaction with mental distress and sexual victimization. Our study involved 538 female and 556 male students from grades 7 to 11. Among the female students, gender dissatisfaction was significantly associated with depression (AOR, 2.04, 95%CI, 1.17–3.58), anxiety (AOR, 2.13, 95%CI, 2.00–2.27), suicidal ideation (AOR, 2.36, 95%CI, 2.02–2.76), sexting victimization (AOR, 1.67, 95%CI, 1.11–2.51), and nonphysical sexual harassment (AOR, 1.72, 95%CI, 1.08–2.76). Among the male students, gender dissatisfaction was significantly associated with oral-–genital contact (AOR, 5.86, 95%CI, 2.74–12.54), attempted sexual assault (AOR, 9.63, 95%CI, 6.91–13.42), and completed sexual assault (AOR, 14.71, 95%CI, 1.16–187.33). Our findings suggest gender dissatisfaction is associated with adolescent mental distress and sexual victimization, underscoring the importance of implementing comprehensive sexual education with gender perspectives in China.

## 1. Introduction

Gender (i.e., the societally shared beliefs that apply to individuals based on their biological sex [1]) can shape human health behaviors and outcomes and is an essential part of comprehensive sexual education (CSE) [2]. For example, there is evidence that masculinity in adolescent boys is associated with the perpetration of violence [3,4], and femininity in adolescent girls suggests an increase in depression [5]. One aspect of gender linked to adverse adolescent health outcomes is gender dissatisfaction. The term currently refers to a discrepancy between an individual’s biological sex and gender identity, and is a risk factor for gender dysphoria [6]. There is little research on gender dissatisfaction, and existing studies have used different measurement items and reported varied prevalence ranging from 2.4% to 6.5% in adolescents [7,8]. Adolescents with gender dissatisfaction often experience increased social stress resulting from discrimination [9], bullying, and harassment victimization [10].

It has been well-documented that social stressors, including harassment and discrimination, have a powerful effect on health [11]. Research on health disparities among marginalized groups indicates that social and minority stress is particularly detrimental to mental health [12]. Discrimination is associated with depression among ethnic minority adolescents [13] and suicidality among sexual minority adolescents [14]. Gender minority youths reported increased sexual victimization [15]. Given that adolescents with gender dissatisfaction often have experiences that overlap heavily with gender minority peers [1,16], they may also be at higher risk for poor mental health and sexual victimization. However, there is little known about the associations between gender dissatisfaction, mental health, and sexual victimization among Chinese adolescents [17,18].

To address this research gap, we conducted a study to describe the prevalence of gender dissatisfaction and examine its association with mental health and sexual victimization among Chinese adolescents. We hypothesized that gender dissatisfaction would be associated with mental health and sexual victimization in both adolescent boys and girls.

## 2. Materials and Methods

### Study Design and Participants

In coordination with local educational authorities, we conducted a cross-sectional study in Minle county, Gansu province. We undertook an anonymous self-reported computer-based survey powered by Sawtooth Ci3 (Sawtooth Software, Provo, UT, USA) in 1 primary school (4 classes in grade 5 and 4 classes in grade 6), 1 junior high school (7 classes in grade 7 and 6 classes in grade 8), and 1 senior high school (6 classes in grade 10 and 6 classes in grade 11) in April 2019.

We included all students attending the three schools during the survey and excluded the students who were absent from school during the survey or refused to participate. Of the 1614 students, 39 declined to participate, leading to a response rate of 91.6%. Given that gender dissatisfaction was only measured in senior and junior students, we excluded all students from primary school, resulting in an analytical sample of 1094.

The present study followed the reporting guidelines in the Strengthening the Reporting of Observational Studies in Epidemiology (STROBE) for cross-sectional studies (eTable).

## 3. Measures

Gender dissatisfaction was measured by the item: “Does your assigned sex bother you (i.e., a boy wants to be a girl or vice versa)?” Response options were “usually”, “often”, “rarely”, and “never”. Students were categorized into two groups of gender dissatisfaction: yes (usually, often) and no (rarely, never), as in previous studies [7,19].

Depression was measured by the Patient Health Questionnaire-2 (PHQ-2), which collects the frequency of little interest or pleasure in doing things or feeling down, depressed, or hopeless over the past two weeks. Anxiety was measured by the Generalized Anxiety Disorder-2 (GAD-2), which collects the frequency of feeling nervous, anxious, or on edge and unable to stop or control worrying over the last 2 weeks. PHQ-2 and GAD-2 were validated in adolescents [20], and the Cronbach’s coefficients of the two scales were 0.7253 and 0.7963, respectively. The score of each item of PHQ-2 and GAD-2 is 0 (never), 1 (several days), 2 (more than half the days), and 3 (almost every day), and each scale ranges from a score of 0 to 6. The recommended cut-points for each scale, when used as screeners for major depressive disorder and generalized anxiety disorder, is a score of 3 or greater [20,21]. Suicidal ideation was measured by the item: “Did you ever seriously consider attempting suicide?” Response options were “yes” and “no.”

We measured a set of 6-item adolescent sexual victimization experiences, including sexting victimization, nonphysical sexual harassment, breast or genital touch, oral–genital contact, attempted sexual assault, and completed sexual assault (Table 1). These items were adopted from our previous study [22].

The demographic characteristics assessed in our study included sex (female or male), ethnicity (Han or non-Han), grade (7, 8, 10, or 11), parental marital status (intact or separated/divorced), parental education attainment (junior high school or lower, senior or vocational high school, or college and above). We also measured relationships with classmates; options included harmonious (good) or nonharmonious (average, bad, or rarely communicate with others).

## 4. Statistical Analysis

Complete case analyses were used. First, we computed descriptive statistics for the demographic characteristics by sex. Differences in the distribution of the demographic characteristics by sex were tested using chi-square (χ^2^) statistics. Second, we estimated the sex-stratified prevalence of gender dissatisfaction, mental distress, and sexual victimization, then we examined the bivariate associations of sex with these variables using χ^2^ statistics. Third, given that gender dissatisfaction, mental distress, and a few indicators of sexual victimization varied by sex, all associations of gender dissatisfaction with mental distress and sexual victimization used were stratified by sex and adjusted for ethnicity, grade, parental marital status, parental higher education attainment, and relationships with classmates in logistic models. We also considered the potential school clustering in estimating the adjusted odds ratios (AORs) and 95% confidence intervals (CIs) for the multivariate associations.

All the data analyses were conducted using Stata/SE 15.1 (StataCorp LLC, College Station, TX, USA), and all statistical tests were considered to be statistically significant if 2-sided *p* < 0.05.

## 5. Results

The descriptive statistics for all demographic characteristics are provided in Table 2. Among the 1094 participants aged 14.42 years (SD = 1.58), 49.18% were girls and 50.82% were boys. Ethnicities were 97.99% Han and 2.01% non-Han. Grade levels were 36.47% in grade 7, 30.62% in grade 8, 16.73% in grade 10, and 16.18% in grade 11. Male students were more prevalent in the lower grades than female students. For parental status, 89.49% reported their parental marital status was intact, and 56.63% reported their parental education attainment was junior high school or lower. In terms of their relationship with classmates, most students (67.73%) reported being harmonious.

Table 3 presents the prevalence of gender dissatisfaction, mental distress, and sexual victimization by sex. The prevalence of gender dissatisfaction was 29.62%, with a much higher prevalence among female students (44.24%) than male students (15.47%). Overall, 9.96% of students reported depressive symptoms, 8.41% reported anxiety disorders, and 19.20% seriously considered attempting suicide. Female students were more likely to report depression and suicidal ideation than male students. With respect to the prevalence of sexual victimization, 17.55% of participants reported sexting victimization (16.36% of girls vs. 18.71% of boys, *p* = 0.307), 23.40% reported nonphysical sexual harassment (20.07% of girls vs. 26.62% of boys, *p* = 0.011), 8.78% reported breast or genital touching (5.02% of girls vs. 12.41% of boys, *p* < 0.001), 1.74% reported oral–genital contact (1.30% of girls vs. 2.16% of boys, *p* = 0.278), 2.83% reported attempted sexual assault (3.53% of girls vs. 2.16% of boys, *p* = 0.171), and 1.65% reported completed sexual assault (1.49% of girls vs. 1.80% of boys, *p* = 0.685).

Table 4 shows the results of the logistic regression, examining associations of gender dissatisfaction with mental distress and sexual victimization among male and female students, respectively. Among female students, gender dissatisfaction was associated with mental distress, including depression (AOR, 2.04; 95%CI, 1.17–3.58; *p* = 0.012), anxiety (AOR, 2.13; 95%CI, 2.00–2.27; *p* < 0.001), and suicidal ideation (AOR, 2.36; 95%CI, 2.02–2.76; *p* < 0.001). We observed similar results among male students for mental distress.

With regard to sexual victimization, gender dissatisfaction was associated with sexting victimization (AOR, 1.67; 95%CI, 1.11–2.51; *p* = 0.013) and nonphysical sexual harassment (AOR, 1.72; 95%CI, 1.08–2.76; *p* = 0.023) among female students. However, among male students, gender dissatisfaction was associated with additional sexual victimization indicators, including oral–genital contact (AOR, 5.86; 95%CI, 2.74–12.54; *p* < 0.001), attempted sexual assault (AOR, 9.63; 95%CI, 6.91–13.42; *p* < 0.001), and completed sexual assault (AOR, 14.71; 95%CI, 1.16–187.33; *p* = 0.038).

## 6. Discussion

In this study, we found that self-reported gender dissatisfaction was prevalent and associated with mental distress and sexual victimization among Chinese adolescents. To the best of our knowledge, this is one of few studies, if any, to examine the associations of gender dissatisfaction with adverse health outcomes in adolescents. Many existing studies focused on its descendants, including gender dysphoria [9,17,23].

Few epidemiological studies on gender dissatisfaction and gender dysphoria among adolescents have been published [7,8]. The prevalence in our study was much higher than findings from other studies. In a Dutch study, the prevalence of gender dysphoria was 2.4% in boys and 3.3% in girls aged 10 years [8]. In another Dutch study with similar measure (“I wish to be of the opposite sex”), the prevalence was 6.5% in adolescents with autism spectrum disorder (ASD) and 3.1% in adolescents without ASD [7]. The apparent differences indicate the differences in the cultural understanding of social gender and a need for more local studies on this topic.

Female students reported a threefold higher prevalence of gender dissatisfaction than male students. Period-related problems [24], including dysmenorrhea and premenstrual syndrome, and masculinity, which is favorably valued in our society, may both have contributed to the sex difference in gender dissatisfaction. We also found that gender dissatisfaction was linked to mental distress, including depression, anxiety, and suicidal ideation, among both adolescent girls and boys. Our findings are inconsistent with those of other studies on gender nonconformity (GNC). These studies suggested similarities and differences in the associations between GNC and mental distress among adolescent girls and boys [1,25]. The minority stress model offers a useful lens through which to understand mental disparities among marginalized populations [26]. Although the model was originally conceptualized for sexual minorities, it has been applied to gender-variant youth [27], including adolescents with gender dissatisfaction in our study. The model specifies that minorities’ exposure to social environmental stressors confers cumulative psychological stress, resulting in potential mental distress [28].

Associations between gender dissatisfaction and sexual victimization varied by sex. Among female students, the prevalence of sexting victimization and nonphysical sexual harassment was higher among students with gender dissatisfaction; other categories of sexual victimization did not vary by gender dissatisfaction. However, among male students, the prevalence of oral–genital contact, attempted sexual assault, and completed sexual assault was higher among students with gender dissatisfaction. In general, adolescents with gender dissatisfaction are at disproportionately higher risks of sexual victimization compared with their peers. The minority stress model also might explain possible causes for sexual victimization vulnerability among adolescents with gender dissatisfaction [29]. Although we did not measure substance use in our study, research shows that gender-minority adolescents are at elevated risk of substance use than their peers [1,30]. Increased substance use may occur as a coping mechanism in response to experiences of social and minority stress [1,31]. The mechanisms to cope with minority stress, according to the minority stress model, may contribute to a higher risk of sexual victimization among marginalized adolescents [15]. In addition, social environments may impact sexual victimization among gender dissatisfaction adolescents. Greater perceived inclusion of sexual- and gender-minority people on campus was associated with significantly lower odds of experiencing sexual assault victimization at college after controlling for sexual orientation, gender identity, race and ethnicity, and year in school in the USA, suggesting that creating an inclusive climate for sexual- and gender-minority individuals may reduce their prevalence of sexual assault [12].

Our findings underscore the need for providing CSE that honors gender equality, free gender expression, and gender diversity during middle school years or earlier, hence having important implications for the implementation of CSE. Our study highlights health disparities in gender dissatisfaction, mental distress, and sexual victimization among adolescents in China. Our finding informs the need to increase awareness of gender diversity among parents, practitioners, and the larger educational community and to create an inclusive climate for gender diversity by strengthening CSE in China, specifically targeting mental distress and sexual victimization among adolescents with gender dissatisfaction. With support from UNESCO and UNFPA, we conducted a large-scale survey on the implementation of CSE from 2014 to 2015 and found that only around 40% of the students received education on sexual orientation and gender identity or reflected on gender inequalities [32]. Developing support systems within schools and families for adolescents with gender dissatisfaction may be an important avenue for improving mental health and reducing sexual victimization in this subpopulation [33]. CSE that is inclusive of discussions about gender, gender identity, and gender expression may be useful in decreasing the stigma for adolescents with gender dissatisfaction [34].

Our findings also have important implications for sexual victimization research, particularly for future studies examining the link between gender dissatisfaction and sexual victimization, which may disproportionately affect adolescent boys in China. Although we conducted several studies on sexting in vocational and college students in Shanghai [35,36,37], no study in China, to the best of our knowledge, has investigated the national prevalence of sexting and its risk factors. More studies are needed to reach a consensus on the definition and measurement of sexting in the Chinese context.

## 7. Limitations

Our study has some limitations. First, our findings apply only to adolescents who attend school, and adolescents with gender dissatisfaction may be more likely to drop out of school or be absent from school [38]. Second, it is possible that students who are willing to report gender dissatisfaction may also be more willing to reveal stigmatized mental health symptoms and sexual victimization experiences, which could inflate the estimated associations between gender dissatisfaction and outcomes studied herein [1]. Third, our study is cross-sectional, which hinders the ability to establish causality. Future longitudinal studies can be designed to better assess a causal relationship between gender dissatisfaction and health outcomes in adolescents. Furthermore, we could not examine how gender dissatisfaction changes over time may affect the outcomes studied herein. Fourth, our findings from adolescents in Minle county may not be generalizable to all Chinese adolescents.

## 8. Conclusions

We reported an increased odds of mental distress and sexual victimization among adolescent girls and boys with gender dissatisfaction in a sample of China. CSE programs that are inclusive of gender diversity in students may help reduce the adverse health outcomes of gender dissatisfaction among adolescents.

## Figures and Tables

**Table 1 children-09-01221-t001:** Questionnaire items and analytic coding of sexual victimization.

Sexual Victimization	Questionnaire Item	Analytic Coding
Receiving a sext	Have you ever received nude or nearly nude pictures or videos via a cell phone?	Yes vs. no
Nonphysical sexual harassment	Have you ever had been told dirty jokes or shown pornographic pictures, publications, supplies, etc.?	Yes vs. no
Breast or genital touch	Has someone ever touched your privates/breasts or forced you to touch their privates/breasts?	Yes vs. no
Oral–genital contact	Has someone ever stimulated your genitals with their mouth or forced you to stimulate their genitals with your mouth?	Yes vs. no
Attempted sexual assault	Has someone ever attempted to have sex with you?	Yes vs. no
Completed sexual assault	Has someone ever forced you to have sex with them?	Yes vs. no

**Table 2 children-09-01221-t002:** Demographic characteristics of all respondents by sex.

Demographic Group	Sex		Chi-Square *p*-Value
	Female, % (*n* = 538)	Male, % (*n* = 556)	
Ethnicity			
Han	98.33	97.66	0.433
Non-Han	1.67	2.34	
Grade			
7	32.90	39.93	0.001
8	29.37	31.83	
10	17.47	16.01	
11	20.26	12.23	
Parental marital status			
Intact	88.48	90.47	0.283
Separated/divorced	11.52	9.53	
Parental education attainment			
Junior high school or lower	55.18	58.03	0.560
Senior or vocational high school	25.42	22.81	
College or above	19.40	19.16	
Relations with classmates			
Harmonious	68.40	67.09	0.642
Nonharmonious	31.60	32.91	

**Table 3 children-09-01221-t003:** The prevalence of gender dissatisfaction, mental distress, and sexual victimization by sex.

Variable of Interest	Sex		Chi-Square *p*-Value
	Female, % (*n* = 538)	Male, % (*n* = 556)	
Exposure			
Gender dissatisfaction	44.24	15.47	<0.001
Mental distress			
Depression	11.90	8.09	0.036
Anxiety	9.85	7.01	0.091
Suicidal ideation	24.54	14.03	<0.001
Sexual victimization			
Receiving a sext	16.36	18.71	0.307
Nonphysical sexual harassment	20.07	26.62	0.011
Breast or genital touch	5.02	12.41	<0.001
Oral–genital contact	1.30	2.16	0.278
Attempted sexual assault	3.53	2.16	0.171
Completed sexual assault	1.49	1.80	0.685

**Table 4 children-09-01221-t004:** Sex-stratified associations of gender dissatisfaction with mental distress and sexual victimization.

Outcome	Gender Dissatisfaction			
	No	Yes		
	%	%	AOR (95%CI)	*p*-Value
Female students				
Mental distress				
Depression	8.67	15.97	2.04 (1.17–3.58)	0.012
Anxiety	7.00	13.45	2.13 (2.00–2.27)	<0.001
Suicidal ideation	17.67	33.19	2.36 (2.02–2.76)	<0.001
Sexual victimization				
Sexting victimization	13.33	20.17	1.67 (1.11–2.51)	0.013
Nonphysical sexual harassment	16.33	24.79	1.72 (1.08–2.76)	0.023
Breast or genital touch	3.67	6.72	1.72 (0.44–6.63)	0.433
Oral–genital contact	0.67	2.10	3.22 (0.50–20.78)	0.219
Attempted sexual assault	3.33	3.78	1.04 (0.59–1.84)	0.886
Completed sexual assault	1.00	2.10	2.28 (0.49–10.50)	0.292
Male students				
Mental distress				
Depression	7.45	11.63	1.65 (1.04–2.62)	0.033
Anxiety	6.60	9.30	1.52 (1.37–1.70)	<0.001
Suicidal ideation	12.77	20.93	2.00 (1.96–2.04)	<0.001
Sexual victimization				
Receiving a sext	16.60	30.23	2.78 (1.66–4.63)	<0.001
Nonphysical sexual harassment	24.47	38.37	2.85 (2.83–2.86)	<0.001
Breast or genital touch	11.28	18.60	1.95 (0.95–3.99)	0.069
Oral–genital contact	1.28	6.98	5.86 (2.74–12.54)	<0.001
Attempted sexual assault	1.28	6.98	9.63 (6.91–13.42)	<0.001
Completed sexual assault	0.64	8.14	14.71 (1.16–187.33)	0.038

Abbreviation: AOR, adjusted odds ratio (adjusted for ethnicity, grade, parental marital status, parental education attainment, and relationships with classmates, with no gender dissatisfaction being the referent group).

## Data Availability

The datasets used and/or analyzed during the current study available from the corresponding author on reasonable request.

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
