# Peer review of "Associations of Gender Dissatisfaction with Adolescent Mental Distress and Sexual Victimization"

_children, 2022, doi:10.3390/children9081221_

Round 1

Reviewer 1 Report

Review comments

The research has dealt with a subject that is rarely spoken or shy to be brought up by the society. In this sense, I think that the study is valuable for the literature.

Although the writing of the study is generally successful, there are some points that need to be corrected and added. After these revisions, we can say that the manuscript will have a better form.

Minor revision

-         - In addition to descriptive information, the introduction should be supported by research results from similar studies in the literature.

-        -  A hypothesis should be added at the end of the introduction.

-          -If there are inclusion or exclusion criteria, they should be added. If there is no criterion, it should be stated that there is no.

-          -"χ2" should be mentioned in a part of the text for clarity. In addition, the expansion should be expressed in writing under the table.

-         - Has power analysis been done? How was the number of participants determined?

-       -   More detailed information about the questionnaires used in the study should be given. Information about its scoring and reliability coefficients should be included.

-       -   It is helpful to use a checklist such as a CONSORT for the standardization of article writing.

-        -  Page 6, line 152, 153; A single reference is given at the end of the sentence, which begins with many existing works. References to the mentioned study should be included as it is stated that there is more than one.

-          -Page 6, line 154, 155; References to these publications should be added to the end of the sentence ending with "few studies have been published".

-        -  At the end of the discussion section, the paragraph is in a form far from the ending sentence. This paragraph and the concluding sentence are similar to the introductory sentence of the discussion section. The subject should be turned into a concluding and concluding sentence.

Reviewer 2 Report

The cross-sectional study explored the relationships between gender dissatisfaction and mental distress, sexual victimization among Chinese adolescents. Due to the large sample size of this study, the sample population is representative. The results showed that gender dissatisfaction was associated with adolescent mental distress and sexual victimization, which has a certain reference value. There are the following suggestions:

1. In the introduction, it is suggested that the definition and prevalence of gender dissatisfaction should be supplemented.

2. In the methods, it is suggested to supplement the Cronbachs coefficients of the scales used.

3. In the results, it is suggested to supplement the analysis results of the relationships between different population indicators and gender dissatisfaction, so as to provide a basis for adjusting these factors when analyzing the relationships between gender dissatisfaction and mental distress, sexual victimization.

4. In the discussion, it is suggested to appropriately supplement the theoretical and reasonable explanation of the relationships between gender dissatisfaction and mental distress, sexual victimization.
